# Assessing the knowledge and attitude towards HIV/AIDS among the general population and health care professionals in MENA region

Ayman Elghazaly[1☯], Nawaf AlSaeed[1☯], Syed Islam[2], Ibtihaj Alsharif[3], Layla Alharbi[3], Tala Al Ashagr[3], Albatoul Alshanifi[4], Reem Alrashoudi[5], Aynaa Alsharidi[4], Abdullah Alhokail[1,6], Qais Dirar[1], Atif Shibl[1], Khaled Al-Kattan[1], Noura Abothneen[7], Maha Al-Mozaini[1,2,5]*

1 College of Medicine, Alfaisal University, Riyadh, Saudi Arabia, 2 Department of Molecular Oncology, King Faisal Specialist Hospital and Research Centre, Riyadh, Saudi Arabia, 3 Department of Infection and Immunity, Immunocompromised Host Section, King Faisal Specialist Hospital and Research Centre, Riyadh, Saudi Arabia, 4 College of Medicine, King Saud University, Riyadh, Saudi Arabia, 5 College of Applied Medical Science, King Saud University, Riyadh, Saudi Arabia, 6 Department of Medicine, King Faisal Specialist Hospital and Research Centre, Riyadh, Saudi Arabia, 7 Department of Women's Health, King Abdullah Foundation, Riyadh, Saudi Arabia

☯ These authors contributed equally to this work.
* mmozaini@kfshrc.edu.sa

## Abstract

### Introduction

Human Immunodeficiency Virus infection continue to represent a global health concern influenced by various social, economic, and cultural factors. The MENA are among the top regions in the world with the fastest-growing HIV epidemic. Thus, adequate knowledge and a positive attitude of people toward HIV/AIDS are of utmost importance to prevent the spreading of the disease. Accordingly, this study aims to evaluate the knowledge and attitude of the public and healthcare population toward HIV/AIDs.

### Methods

A cross-sectional analysis was conducted among residents within our population from October 2018 until August 2019. An anonymous online questionnaire was used to investigate the population's demographic characteristics, HIV/AIDS-related knowledge, and attitudes toward HIV-infected patients. Participants completed a 40-item questionnaire designed to measure their knowledge and attitude toward HIV/AIDS. The data was collected via surveys, administered through electronic tablets to the participants at public places (n = 5,757) and through an online version of the questionnaire on Google Forms (n = 2500), which was sent through social media platforms. Descriptive statistics were used to analyse the data using the R-statistical software program.

**Data Availability Statement:** All relevant data are within the paper and its Supporting Information files.

**Funding:** The author(s) received no specific funding for this work.

**Competing interests:** The authors have declared that no competing interests exist.

**Abbreviations:** MENA, Middle Eastern and North Africa; HCW, Health Care Worker; NHW, Non-Health Care Worker.

## Results

A total of 8,257 participants were included in our analysis. Saudi Arabian citizens represented 79% of the participants, while participants from the MENA countries represented 11.7% and 3% from the other Gulf Cooperation Council countries. Fifty-nine (59%) knew that HIV is a contagious infection, and 13.8% were unaware that HIV could be transmitted sexually. A few healthcare professionals reported negative attitudes toward HIV infected patients. Many risk factors, including age, gender, nationality, and education, significantly affected the knowledge and attitude scores. In this survey, we found that social media is the primary source of participants' information.

## Conclusions

Overall correct knowledge score of individuals about HIV/AIDS was relatively low. This study showed that the general population was knowledgeable to a certain degree about HIV/AIDS and its modes of transmission. Nevertheless, they lack a detailed understanding of the disease's nature, modes of transmission, and existing treatment. Policymakers in the region should further eliminate social discrimination and stigma in HIV-infected patients.

## Introduction

It has been four decades since the discovery of the Human Immunodeficiency Virus (HIV) and its ongoing work to reduce new infections. However, Acquired Immunodeficiency Syndrome (AIDS) still represents one of the leading global health concerns, influenced by various social, economic, and cultural factors. According to a recent report by the World Health Organization (WHO) and the joint United Nations Program on HIV/AIDS (UNAIDS), 38 million people are currently infected with HIV worldwide [1]. There were approximately 1.5 million newly infected globally in 2021 alone, indicating that the disease burden is rising. Recent reports show that there has been steady progress in eastern and southern Africa, where new HIV infections have reduced by 38% since 2010.2 In contrast, new HIV infections have risen in the Middle East and North Africa (MENA) regions by 22% [2].

The Saudi Arabian Ministry of Health documented a total of 22,952 cases of HIV infections between 1984 and 2015 [3]. A more recent study showed that between the years 2000–2009, there was an increase in the annual incidence of HIV from 125 to 481 cases [4]. However, these numbers may not reflect the incidence of HIV infection due to a lack of knowledge about the disease and the hesitancy to perform screening tests, particularly for those at high risk. In addition to the adolescents and adults in Saudi Arabia who are relatively well educated, many still have insufficient information about HIV/AIDS. A limited number of studies have been conducted to assess students' knowledge and attitudes toward HIV/AIDS in Saudi Arabia [5]. Also, few studies in Saudi Arabia indicated that the general population has insufficient knowledge of HIV/AIDS. However, all these mentioned studies have several limitations, such as the small sample size and data collection from specific regions. Thus cannot be generalized to the whole regions.

Therefore, this study aimed to examine and assess the knowledge and attitude toward HIV/AIDS of the general population through a comprehensive survey to plan and develop an adequate HIV/AIDS awareness compiagn in the region. We report the differences and scores of HIV/AIDS knowledge and attitudes towards people living with the disease. Our study can help

policymakers design comprehensive HIV/AIDS health education plans and arrange adequate national awareness campaign programs.

## Methods

### Study population and data collection instrument

A cross-sectional analysis was conducted among residents aged 15 and above in Saudi Arabia from October 2018 until August 2019. The study size was 8,852 participants and within the 99% confidence interval of Saudi Arabia's total population of 34 million. Initially, 5,757 surveys were administered by volunteers anonymously after they consented to participate in the study on electronic gazette to the participants at shopping malls and KFSH&RC. Then, additional 2,500 surveys were completed through an online version of the questionnaire on Google Forms™. The online version of the survey in a form of Google Forms link was disseminated using different social media platforms' posts for instance the survey was posted on the official Twitter page of King Faisal Specialist Hospital & Research center. Furthermore, the questionnaire's online version link was shared by data collection volunteers using their social media platforms such as WhatsApp.

### Survey design

A 40-item questionnaire was designed and prepared based on a literature review. Before administration, the questionnaire was piloted on a random sample of 50 participants to augment the study's internal validity. Based on the pilot survey results, some questions were rephrased and modified. The questionnaire was made available in both English and Arabic languages. Participants self-completed a structured questionnaire that consisted of 40 closed-ended questions, all of which were a single response.

The questionnaire is divided into four different sections: (i) data on participants' social and demographic characteristics, including gender, age, nationality, and marital status. (ii) 15 questions to measure the knowledge and awareness of the general population about HIV/AIDS. (iii) 6 questions to measure the attitude and behavior of medical professionals toward HIV/AIDS patients. (iv) 9 questions to assess the general population's attitude and stigma score toward HIV/AIDS patients. We also had a couple of questions about media influence, as summarized in Table 1.

### Managing missing data

Initially, there was an incident of missing data within the Arabic version for the first 500 participants enrolled via social media. Therefore, we extrapolated the available data to overcome this challenge and ensure that the data used passed the test. We did this using the relationship between the variables in the dataset. Doing this gave us a rich and reliable dataset and helped the researchers remove the uncertainties or possible biases that might arise from the missing data.

### Statistical methodology

Pearson correlation nonparametric analysis ANOVA was used, then the R-statistical program was utilized in data grouping and coding. Each group of participants was analyzed and compared. The association between each risk factor and knowledge prevalence was assessed by the odds ratio (OR) and associated 95% confidence interval adjusted for the stratifying variables. All risk factors and stratifying variables were combined into a multivariate logistic regression

**Table 1. List of questions for the study survey.**

| Question Number | Category |
|---|---|
| | **General Social and Demographics** |
| 1 | Gender (Male, Female) |
| 2 | Age (15–19, 20–29, 30–39, 40–49, & 50 <) |
| 3 | Nationality (Saudi Arabia, GCC countries, Middle East and North Africa, Others) |
| 4 | Region of Residence (Northern Province, Southern Province, Eastern Province, Western Province, Central Province) |
| 5 | Marital Status (Single, Married, Previously married "*Divorced*, *widow*") |
| 6 | Educational Level (Primary Education, High School, Undergraduate Level, or Higher Education) |
| 7 | Monthly Income (<5000, 6000–10,000, 10,000–25,000, 25000<) |
| 8 | Occupation (Student, Employed, Unemployed) |
| | **Media Influence** |
| 9 | How often do you use the three mass media sources- Television, radio, and newspaper? (Almost every day, At least once a week, Less than once a week, and Not at all) |
| 10 | How often do you use social media sources? (Almost every day, At least once a week, Less than once a week, and Not at all) |
| | **Knowledge & Awareness of HIV/AIDS** |
| 11 | Have you ever heard of HIV or the illness AIDS? |
| 12 | What was your source of knowledge regarding HIV or AIDS? (Mass Media, Social Media, Brochures, Seminars/Booths, Infected individuals, others) |
| 13 | Do you know what the difference between HIV infection and AIDS is? (Yes, No) |
| 14 | Is HIV contagious? (Yes, No, I do not know) |
| 15 | Can HIV be transmitted through daily contacts, such as sharing public bathrooms, food, or utensils? (Yes, No, I do not know) |
| 16 | Can a person get HIV through contact with saliva, tears, urine, or sweat? (Yes, No, I do not know) |
| 17 | Does coughing and sneezing spread HIV? (Yes, No, I do not know) |
| 18 | Can a person get HIV by blood transfusion? (Yes, No, I do not know) |
| 19 | Can HIV be sexually transmitted from one person to another? (Yes, No, I do not know) |
| 20 | Is AIDS a disease of homosexuals or bisexuals **only**? (Yes, No, I do not know) |
| 21 | Can a healthy-looking person have HIV/AIDS? (Yes, No, I do not know) |
| 22 | Do people infect with HIV quickly develop signs of severe disease? (Yes, No, I do not know) |
| 23 | Once infected with HIV, can a patient transfer the virus at all stages of life? (Yes, No, I do not know) |
| 24 | Is AIDS curable? (Yes, No, I do not know) |
| 25 | Is prevention the most significant cure for HIV? (Yes, No, I do not know) |
| | **Stigma and Attitude towards HIV/AIDS in HCW** |
| 26 | Do you study or work in healthcare? (Yes, No) |
| 27 | Will you treat a patient infected with HIV? (Yes, No) |
| 28 | Would you provide the same quality of care to HIV-positive patients that you provide to other patients? (Yes, No) |
| 29 | Would you be willing to do a physical examination of a known HIV-positive patient? (Yes, No) |
| 30 | Do you dislike having physical contact with HIV/AIDS patients? (Yes, No) |
| 31 | Would you interact with HIV-positive patients just like other patients? (Yes, No) |
| | **Stigma and Attitude towards HIV/AIDS in General Population** |
| 32 | Do you feel afraid of persons living with HIV/AIDS? (Yes, No) |
| 33 | Did the people who got HIV through sex or drug use have what they deserved? (Yes, No) |
| 34 | Is AIDS a punishment for inappropriate behavior? (Yes, No) |
| 35 | Should HIV-positive patients be blamed for their condition? (Yes, No) |

(*Continued*)

**Table 1.** (Continued)

| Question Number | Category |
|---|---|
| 36 | Would you feel ashamed if someone you know got HIV? (Yes, No) |
| 37 | Will your attitude change towards your colleagues if they contract HIV? (Yes, No) |
| 38 | Will you spend time with your friend if you come to know that he has HIV? (Yes, No) |
| 39 | Do you approve of the marriage of an HIV-positive individual to a healthy HIV-negative individual? (Yes, No) |
| 40 | Do you approve of the marriage of two HIV-positive individuals? (Yes, No) |

Note: HCW = Healthcare Workers

analysis to account for potential confounders. Results were statistically significant at the 5% critical levels (p<0.05). The calculation was done using R-statistical software packages.

## Ethics

Ethical approval was granted by the office of the research affairs committee at KFSH&RC (RAC#2181 184). Participation in the study was voluntary, and all participants were informed about the study objectives and were assured that their responses and participation would be anonymous. Some participants were less than 18 years, 12% of our study participant. The consent of their parents/guardians were obtained before they were interviewed The research affairs committee has approved wavier need of informed consent from participants. Only verbal consent was taken from the participants by the data collection team.

## Results

### Participants' socio-demographic characteristics

A total of 8,257 participants were included in our survey analysis. Most respondents (97·8%; n = 8075) have heard about the two terminologies, either HIV or AIDS. Female participants in our survey were more (63·5%; n = 5243) than male participants (36·5%; n = 3014). About half of our study participants (49·3%; n = 4075) were over 30 years old. Seventy-nine percent were Saudi Arabian citizens (n = 6522) residing mainly in the kingdom's capital, Riyadh, while 61.8% (n = 4021) were within the central region. The remaining participants were from MENA countries, 11.7% (n = 963), followed by other GCC countries, 3% (n = 245).

Fifty percent (n = 4162) of the respondents were single, and 45·2% (n = 3731) were married. Of the remainder, 1·1% were divorced, separated, or widowed, and 1% did not specify their marital status. Of the 7,167 surveys which contained the question about the educational level of the participant, 199 (2·8%) participants had a primary school degree or below, 1,237 (17·3%) had a secondary or high school degree, 4,478 (62·5%) had an undergraduate degree, and 1,253 (17·5%) had a postgraduate degree. One-third of the participants are studying or working in the medical field, 2723 (33%). These overall general characteristics are summarized in Table 2.

### Participants' general knowledge of HIV/AIDS

Many participants (97·8%; n = 8075) have heard about HIV or AIDS. Among the 12 questions that assess the overall HIV/AIDS Knowledge of viral mode of transmission and treatment. Most participants, 7,222 (87·5%), knew that blood transfusion was the main route of viral transmission. However, a low number of respondents, 36·4% (n = 3007), knew that viral

**Table 2. General characteristics of our study population.**

| Characteristics | | Total Number | Percentage (%) |
|---|---|---|---|
| **Gender** | Male | 3014 | 36·5% |
| | Female | 5243 | 63·5% |
| **Age** | 15–19 | 1024 | 12·4% |
| | 20–29 | 3158 | 38·2% |
| | 30–39 | 1704 | 20·6% |
| | 40–49 | 1243 | 15·1% |
| | Above 50 | 1128 | 13·7% |
| **Nationality** | Saudi Arabian | 6522 | 79·0% |
| | GCC | 245 | 3·0% |
| | MENA | 963 | 11·7% |
| | Others | 527 | 6·4% |
| **Region** | Central | 4021 | 61·8% |
| | Eastern | 863 | 13·3% |
| | Northern | 322 | 4·9% |
| | Western | 944 | 14·5% |
| | Southern | 357 | 5·5% |
| **Marital Status** | Single | 4162 | 50·4% |
| | Married | 3731 | 45·2% |
| | Previously married | 364 | 4·4% |
| **Education** | Primary | 199 | 2·8% |
| | Secondary | 1237 | 17·3% |
| | Under-Graduate | 4478 | 62·5% |
| | Postgraduate | 1253 | 17·5% |
| **Monthly Income** | Less than 1000 | 3160 | 38·3% |
| | 1000–4999 | 2927 | 35·4% |
| | 5000–10000 | 1232 | 14·9% |
| | More than 10000 | 938 | 11·4% |
| **Occupation** | Student | 2829 | 34·3% |
| | Employed | 4846 | 58·7% |
| | Self-employed | 454 | 5·5% |
| | Unemployed | 128 | 1·6% |
| **Heard of HIV or AIDS** | No | 181 | 2·2% |
| | Yes | 8075 | 97·8% |
| **Studying or working in a healthcare institute?** | No | 5533 | 67·0% |
| | Yes | 2723 | 33·0% |

transmission could occur at any stage of the disease. Of note, knowledge about the various transmission methods was variable, as shown in S1 Table in S1 File. Nonetheless, most participants answered correctly about the misconception of HIV transmission mode. However, 13·8% were unaware that HIV could be transmitted sexually. In addition, more than 85% knew that HIV infection could be preventable, and 50·9% lacked the knowledge of available HIV treatment and cure strategies. Moreover, only 59·3% (n = 4895) knew that HIV is a contagious infection. Finally, there was no statistically significant difference between the knowledge of male and female participants.

## The attitudes and perceptions of healthcare professionals to HIV/AIDS

Participants linked to healthcare professionals comprise 33% of the cohort (n = 2723). Our survey analysis indicated that the healthcare professionals believed in providing equal patient care and attention to those HIV-infected individuals like any other patients. Nevertheless, some of them expressed hesitancy to interact directly with patients. For example, one-third of the healthcare professional participants (n = 892) reported preferring not to have physical contact with HIV-infected patients. Also, 595 (21·9%) respondents stated that they would interact with their HIV-infected patients differently than other patients, as summarized in S2 Table in S1 File. However, healthcare professionals have shown significantly better attitudes and perceptions about HIV-infected patients than the general population.

Moreover, about 17·1% of healthcare professionals believed that HIV/AIDS is a punishment for inappropriate behavior, while 30·6% of the general population shared the same misconception (P < 0.001). Both groups, healthcare professionals and the general population, endorsed that their attitude would change toward their colleagues if they contracted HIV infection, 21·4% and 36·3%, respectively, P < 0.001, as summarized in S3 Table in S1 File. Furthermore, most healthcare professionals and the general population disapprove of the marriage of an HIV-positive individual to an HIV-negative individual, 85·9% vs. 88·6%, respectively, with a significance of P < 0·001.

## Knowledge score of HIV/AIDS among the different groups in our population

We assessed the difference in HIV/AIDS knowledge within different precipitant groups, including age, nationality, residency region, marital status, education level, income, occupation, and media source, as summarized in S4 Table in S1 File. Age categories of < 20 and > 50 years showed lower knowledge scores than other age groups, with a significant p-value of < 0·001. Despite the low number of participants from other GCC residents, 3% showed higher knowledge scores than Saudi and other nationalities, with a significant P value of < 0·001. Encouraging data shows that individuals with postgraduate education levels had better HIV/AIDS knowledge than undergraduate and school student degrees. However, occupation status was not linked to any knowledge difference, with a P value of 0·175.

Regarding marital status, single patients had the highest general HIV knowledge score compared to married or previously married participants, with a significant P value < 0·05. However, there was no difference in the HIV mode of transmission knowledge score (p = 0·27). However, around 85% knew that HIV could be sexually transmitted, decreasing the HIV transmission risk for unmarried and married patients.

Since Saudi Arabia is a large country with different cultures and tribes, our study provided extra insight into the different regions around the kingdom. The western and central regions showed the highest and lowest knowledge scores, respectively, as shown in S4 Table in S1 File [p = 0·053]. However, there was no significant difference in the HIV mode of transmission knowledge score [p = 0·12]. These data indicate the need for awareness campaigns all over the kingdom.

The mean knowledge score between males and females was similar, with a statistically insignificant difference. In contrast, a notable gap in HIV knowledge scores was noted between healthcare workers and non-healthcare workers. "No" represents the non-healthcare workers, while "Yes" represents the healthcare workers, as seen in (Fig 1).

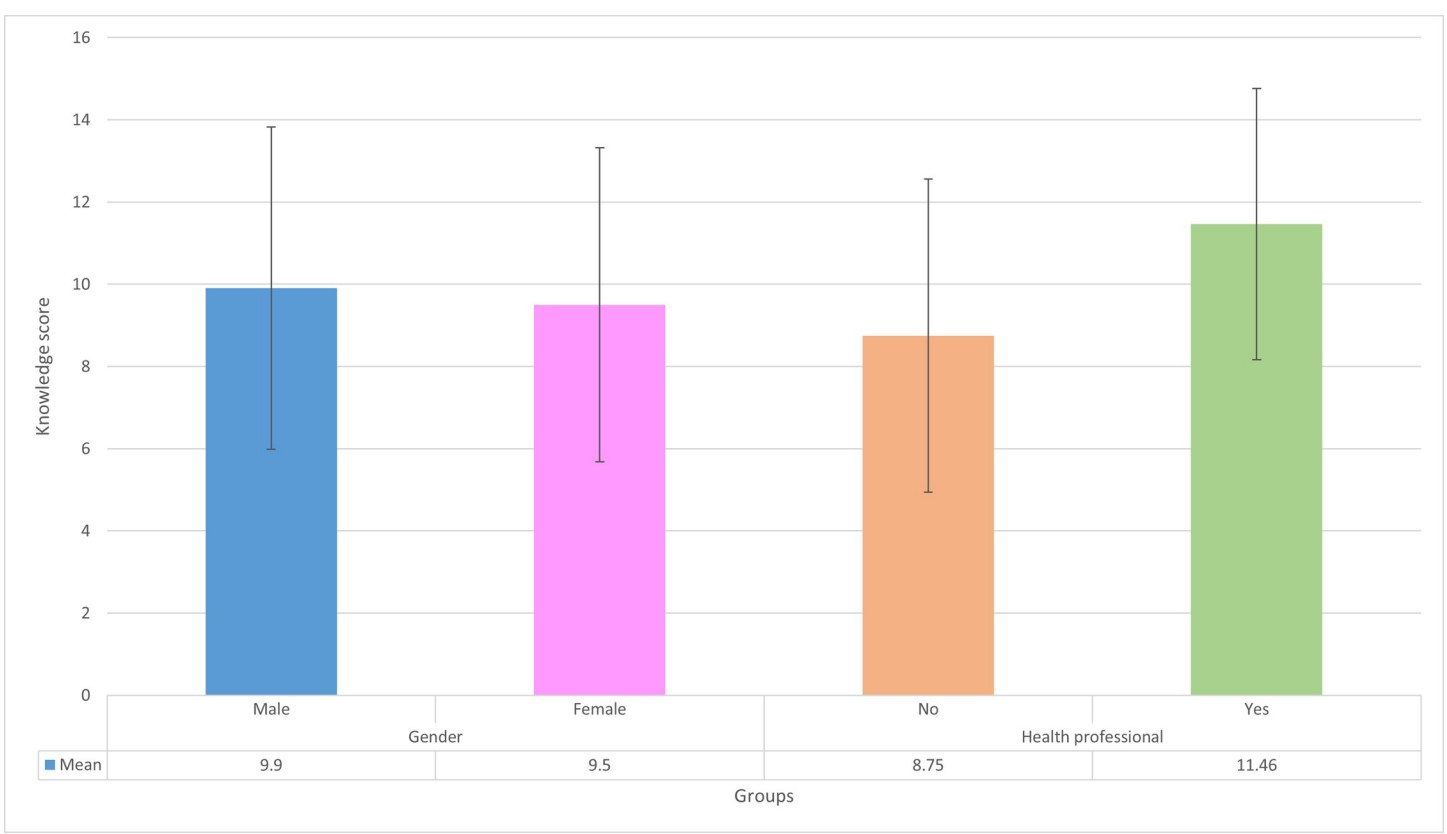

**Fig 1. HIV knowledge score among gender and HCW VS NHW.** The histogram represents the mean HIV/AIDS knowledge score between males (9.9) and females (9.5), which shows similar scoring and is not statistically significant. In contrast, there is a notable gap in HIV/AIDS knowledge scores between healthcare and non-healthcare workers, 8.75 versus 11.46 respectively. "No" represents the non-healthcare workers, while "Yes" represents the healthcare workers.

### Knowledge score of HIV/AIDS prevalence and mode of transmission by demography

The knowledge score of HIV/AIDS prevalence and transmission mode is summarized in S4 Table in S1 File. Most of the mode transmission was answered correctly by participants in the country's central region, including Riyadh, which might be attributed to the high number of participants from this region. Our observations show that younger generations, less than 40, were more familiar with the mode of viral transmission ([$p < 0.05$]). In addition, the undergraduate university students were more knowledgeable of viral transmission, with a significant p-value of ([$< 0.01$]). Another supersizing finding was that low-income participants answered the question correctly, with a significant p-value of ([$< 0.05$]).

### Overall HIV/AIDS knowledge and stigma score per group

We also investigated the overall HIV/AIDS knowledge and stigma per group. Gender, marital status, education, and income were used as the categorizations. On question 14, 38·7 percent of males and 41·9 percent of females revealed as not having knowledge and stigma about HIV/AIDS, compared to 61·3 percent and 58·1 percent [$< 0.04$]. A considerable percentage of married couples (57·7%), single (61·2%), and previously married (54·1%) were also knowledgeable about HIV/AIDS and stigma as compared to those who do not have any knowledge [$< 0.01$]. The study also revealed that a considerable percentage of participants with an education

(primary, secondary, undergraduate, or postgraduate) are more knowledgeable about HIV/AIDS and issues with stigma than those without education. In addition, the number of those aware and knowledgeable tended to increase with the level of education, with those with primary school education at 51·8% and those with postgraduate education at 68·2%. Lastly, the study wanted to determine the level of awareness, knowledge, and stigma of different income categories. There are slight changes in those who responded with "Yes" over the different income brackets. The results show that gender, marital status, and education have a role in knowledge and stigma compared to income level.

## HIV/AIDS knowledge through media as a source of information

Identifying which media sources are frequently used in our community will help us develop and establish future HIV/AIDS awareness campaigns. For this reason, we collected data on survey questions reflecting the types of media frequently used and their effect on participants' knowledge. For example, nearly 80% of participants use social media, while less than 50% are reluctant to use mass media daily. However, knowledge among participants who use mass media and social media is near-equal. The results from the study reveal that social media is not only the most famous avenue for communicating HIV/AIDS campaigns, but it also has the highest coverage in the region. The study also revealed dwindling usage of mass media, as shown in S6 Table in S1 File.

## Discussion

To our knowledge, this is the first comprehensive study to assess the knowledge, attitudes, and stigma toward HIV/AIDS in MENA region. In addition, this is the largest national survey conducted and targeted the general population and healthcare workers in the region since UNAIDS 90-90-90 program was initiated and concluded.

In 2014, The UNAIDS set a 90-90-90 global target to limit the AIDS epidemic by the year 2020 by ensuring that 90% of all people living with HIV should know their HIV status, 90% of all people diagnosed with HIV infection should receive sustained antiretroviral treatment, and 90% of all patients who receive antiretroviral therapy should have viral suppression. This target was achieved in a few cities worldwide, but it remains a challenge in many countries, especially in the gulf and MENA region. Therefore, it is necessary to assess the knowledge and attitude of our population and healthcare workers and help promote testing and treatment through HIV/AIDS awareness campaigns.

The UNAIDS target starts with recognizing the disease, which means the availability of screening centers and the level of knowledge and disease awareness within a targeted community. The awareness should motivate community members to pursue HIV testing, especially those in high-risk groups. For this reason, we have surveyed to assess the population's knowledge and attitude toward HIV/AIDS. The data collected provide a more in-depth understanding of the current national situation. In addition, they will allow for a more tailored approach to improving awareness and reducing the region's stigma associated with HIV/AIDS. This survey was conducted at the end of 2019 while approaching 2020, which was the UNAIDS targeted year for significantly limiting the AIDS epidemic that is yet not achieved in the MENA region. We believe this study will aid agencies in Saudi Arabia and other MENA countries in conducting different programs to close the knowledge gaps we have found in this survey.

Most of our respondents, 97·8%, have heard about HIV/AIDS. However, more than half (53%) did not know the difference between HIV and AIDS. Furthermore, considering that most participants (>70%) were at a sexually active age, almost 40·7% of participants did not realize that HIV is a public health concern. Additionally, around 63% were not aware that HIV

could be transmitted at any stage of the disease. These results represent a poor knowledge of HIV transmission risk, which is enough for many sexually active men and women in our society to neglect HIV testing and the use of protection despite being at risk for it.

One of the main challenges an individual living with HIV/AIDS in Saudi Arabia and other MENA countries may face is stigma and discrimination. An example of such associated stigma is seen in the attitude responses in S5 Table in S1 File, where 57% of respondents, including HCW and NHW, feel afraid of people living with HIV/AIDS. Indeed, one-third thought of changing their attitude and would not spend time with their HIV-positive colleagues. To our surprise, only 54% supported marriage between HIV-positive couples. These results show that in certain aspects linked to incorrect knowledge about HIV, there was a modest negative attitude toward people living with HIV in region regardless of the professional background of the respondent. The main reason for this behavior is a lack of knowledge about the mode of transmission and available modern therapies to control HIV and prevent HIV transmission between discordant couples and mother-to-child. No doubt, such an attitude compromises the quality of life for people who live with HIV, affecting the psycho-social elements of HIV-positive patients. These negative attitude factors may severely impact the second and third targets of the UNAIDS' 90-90-90 plan.

For many reasons, the healthcare provider's sufficient knowledge and positive attitude towards HIV/AIDS and those afflicted by it is of utmost importance. First, healthcare providers can participate in the initiation of awareness campaigns and provide correct information on preventative measures. Second, healthcare providers can help provide accurate information directly to patients they see as high-risk and refer them to screening programs. Finally, the healthcare provider's attitude toward HIV patients may negatively impact their care plan. In 2009, Memish et al [6], assessed the knowledge and attitude of Saudi Arabian doctors and found poor knowledge of HIV, mainly modes of transmission [7]. Our survey showed that most healthcare providers have a generally positive attitude towards HIV-positive patients. However, it is far from the acceptable zero percent negative attitude required for fully standardized proper care and management. These results justify starting healthcare worker-oriented programs to improve their attitudes and solidify their general knowledge [8].

We found that many risk factors were associated with significant knowledge differences within our participant's groups, including the lower level of awareness among extreme age (< 20 and > 50 years old), Saudi nationality, low income, married, undergraduate, and school students. There is not much difference in the educational or living standards, but overall, there is a trend that the higher the income level, the higher the awareness rate. These results reflect which groups need to be targeted and have more tailored awareness programs than others. Nevertheless, an awareness campaign is indicated in all the Kingdom regions as they have identical knowledge scores.

We propose a new focus on social media and mass media in spreading awareness of HIV/AIDS. As demonstrated in (Fig 2), most participants used either or both means of media frequently. Despite this, many participants have a notable knowledge gap about HIV/AIDS. Therefore, it could be inferred that mass media and social media are underutilized targets for awareness programs with the potential to close the gap.

Many studies worldwide have shown that awareness and educational intervention can effectively improve people's knowledge about HIV/AIDs, increase the student population's awareness of HIV/AIDS-related knowledge [9], and improve people's discrimination against AIDS groups [10]. In today's internet age, we can adopt various new ways to educate our population on AIDS prevention and treatment; compared with traditional book-based teaching, these new educational methods integrate knowledge into real life and present it to people

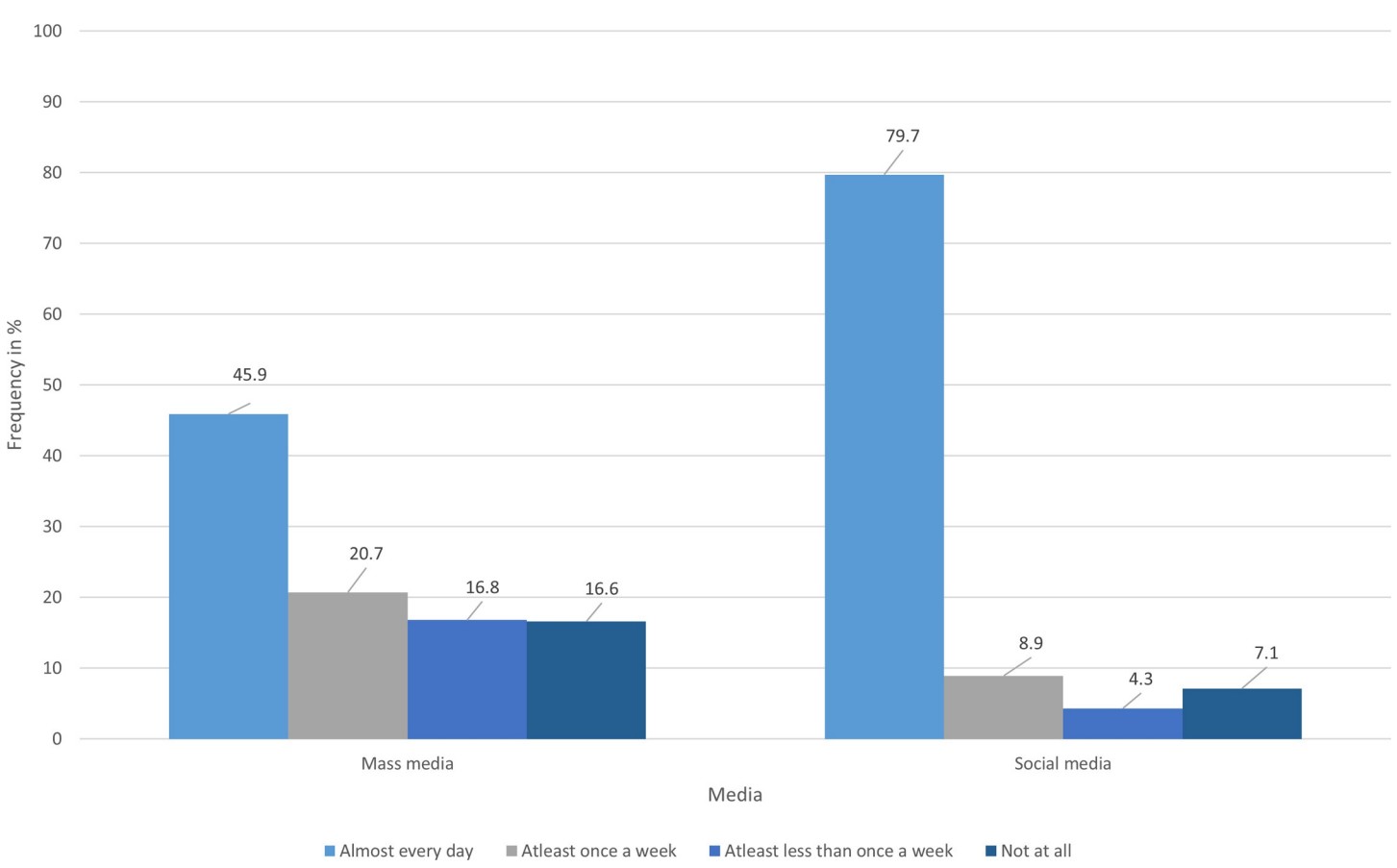

**Fig 2. Frequency of utilization of mass media and social media among the study participants.** A histogram that Identifies which media sources are frequently used in our community. The majority of our participants, 80%, use social media daily, and less than 50% of participants are reluctant to use mass media daily.

more vividly. It is easier to arouse the interest of young people and thus obtains better educational effects.

Our survey showed limited awareness of HIV/AIDS in our community. Therefore, there is an urgent need to provide sufficient information on HIV/AIDS to the community and have public health policymakers endorse national HIV awareness programs. This neglect could be due to the low national prevalence of the disease, sensitivity to the topic, and cultural barriers all play a role. However, considering the increased prevalence of HIV/AIDS in the MENA region, preventive education is an integral and essential part of an effective and comprehensive combination HIV prevention program. Hence, sufficient knowledge and education programs should be mainstreamed across the community, especially among young people, as in universities and schools to prevent new HIV infections.

## Limitations

Our study has couple of limitations, one which is the geographical representation of the participants. The study data was collected from all different MENA regions. However, the majority of our participants were Saudi Arabia. Furthermore, most of the in-person survey data collection was done in Riyadh due to the logistical difficulty with the 'Data Collection' volunteers. This limitation was worked on by providing an online version of the questionnaire in the form of a link that was shared on social media to maximize the geographical reach of our survey to

all MENA regions. In addition, the administrative difficulties faced by a few institutions led to the location's decline to participate in conducting the study on their premises, thus decreasing the potential number of participants in the study. Nonetheless, upon conducting the study, there was great enthusiasm within the community at every step to participate. Despite the limitations we referred to above, the results of this study can still provide references for policymakers to implement HIV/AIDs awareness campaigns.

## Conclusions

We have highlighted the most extensive study so far on the current level of knowledge of the general population about HIV/AIDS in MENA. The general attitude was positive, which should be a key asset in spreading awareness about HIV/AIDS. However, the attitude of healthcare workers needs more improvement to provide the best amount of care to the HIV-afflicted. Therefore, we need to readdress these major elements and work with them to reduce the gaps in knowledge, decrease stigmatizing attitudes towards people living with HIV/AIDS, and reach the UNAIDS zero-zero-zero plan in Saudi Arabia. We suggest that social media platforms are the best place to achieve these goals in our community.

## Supporting information

**S1 File. HIV survey supplementary material.**
(DOCX)

**S1 Data. Brief description of file content.**
(XLSX)

## Acknowledgments

We like to thank everyone who participated in the study. We also thank all medical students that helped in conducting this study and making this research possible. Furthermore, we are grateful to all individuals and all the different sectors who have assisted in the smooth progress of this research. Furthermore, the support of the Research Centre administration at King Faisal Specialist Hospital & Research Centre is highly appreciated. Finally, we would like to thank the non-profit organization that supports women living with HIV/AIDs; for their continuous support https://www.nismh.org.

## Author Contributions

**Conceptualization:** Maha Al-Mozaini.

**Data curation:** Ayman Elghazaly, Nawaf AlSaeed, Tala Al Ashagr, Albatoul Alshanifi.

**Formal analysis:** Nawaf AlSaeed, Syed Islam, Ibtihaj Alsharif, Layla Alharbi, Tala Al Ashagr, Albatoul Alshanifi, Maha Al-Mozaini.

**Investigation:** Syed Islam, Ibtihaj Alsharif, Layla Alharbi, Maha Al-Mozaini.

**Methodology:** Ayman Elghazaly, Nawaf AlSaeed, Tala Al Ashagr, Albatoul Alshanifi, Maha Al-Mozaini.

**Writing – original draft:** Ayman Elghazaly, Nawaf AlSaeed, Tala Al Ashagr, Albatoul Alshanifi, Maha Al-Mozaini.

**Writing – review & editing:** Syed Islam, Ibtihaj Alsharif, Layla Alharbi, Reem Alrashoudi, Aynaa Alsharidi, Abdullah Alhokail, Qais Dirar, Atif Shibl, Khaled Al-Kattan, Noura Abothneen.

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
