## [Decision Letter · Decision Letter 0]

1 Mar 2023

PONE-D-22-34156Assessing the Knowledge and Attitude Towards HIV/AIDS among the General population and Health Care Professionals in MENA regionPLOS ONE

Dear Dr. AlMozaini,

Thank you for submitting your manuscript to PLOS ONE. After careful consideration, we feel that it has merit but does not fully meet PLOS ONE’s publication criteria as it currently stands. Therefore, we invite you to submit a revised version of the manuscript that addresses the points raised during the review process.

We look forward to receiving your revised manuscript.

Kind regards,

Frank Kyei-Arthur, Ph.D.

Academic Editor

PLOS ONE

Journal Requirements:

2. You indicated that you had ethical approval for your study. In your Methods section, please ensure you have also stated whether you obtained consent from parents or guardians of the minors included in the study or whether the research ethics committee or IRB specifically waived the need for their consent

3. Please update the Figures captions, all figures captions as Figure 1(Figure 1. HIV Knowledge Score Among Gender and HCW VS NHW, Figure 1. Frequency of Utilization of Mass Media and Social Media among the Study Participants)

Additional Editor Comments:

The author should address the comments of the reviewers, especially reviewer 2. The author should pay attention to his/her sample collection protocol, data analysis and results.

Reviewers' comments:

Reviewer's Responses to Questions

**Comments to the Author**

1. Is the manuscript technically sound, and do the data support the conclusions?

Reviewer #1: Yes

Reviewer #2: Partly

2. Has the statistical analysis been performed appropriately and rigorously? 

Reviewer #1: Yes

Reviewer #2: N/A

3. Have the authors made all data underlying the findings in their manuscript fully available?

Reviewer #1: Yes

Reviewer #2: Yes

4. Is the manuscript presented in an intelligible fashion and written in standard English?

Reviewer #1: Yes

Reviewer #2: Yes

5. Review Comments to the Author

Reviewer #1: This is a very important manuscript. It highlights the general public knowledge on HIV/AIDS. most importantly its highlight the perception of health workers towards HIV patients. The attitude of healthcare workers needs more improvement to provide the best amount of care to the HIV-afflicted.

Reviewer #2: This is a good study, unfortunately dataset and results is not unique. The paper has many shortcomings in regards to data analyses and text. In my opinion, However, it need to more clarify regarding to the protocol of sample collection and protocol which is used to online survey.In addition, results has not been used to its full text. I think need more in depth analysis of the data. I also suggest siting more relevant and new literatures.

6. PLOS authors have the option to publish the peer review history of their article (what does this mean?). If published, this will include your full peer review and any attached files.

Reviewer #1: **Yes: **Adwoa Asante-Poku

Reviewer #2: No

---

## [Author Response · Author response to Decision Letter 0]

1 May 2023

Dear Editor in Chief March 25, 2023

PLOS ONE 

We would like to thank the reviewers and the Editorial Office again for their time and their critical review of our manuscript. We are pleased to inform you that all the comments raised by the reviewers were taken carefully responded point-by-point. Please find below our point-by-point responses. All the changes in the attached revised manuscript are indicated in track change. 

We thank you again for the opportunity to resubmit our paper.

Yours faithfully

Maha Al-Mozaini, PhD

Reviewer comments:

1. You indicated that you had ethical approval for your study. In your Methods section, please ensure you have also stated whether you obtained consent from parents or guardians of the minors included in the study or whether the research ethics committee or IRB specifically waived the need for their consent.

Response: we have included this statement at the Ethics part of the manuscript.

2. Please update the Figures captions, all figures’ captions as Figure 1(Figure 1. HIV Knowledge Score Among Gender and HCW VS NHW, Figure 1. Frequency of Utilization of Mass Media and social media among the Study Participants)

Response: amended 

Reviewer comments

1. Is the manuscript technically sound, and do the data support the conclusions?

Reviewer #1: Yes

Reviewer #2: Partly

Response 1: On behalf of all the coauthors and myself, we appreciate this positive feedback. 

2. Has the statistical analysis been performed appropriately and rigorously? 

Reviewer #1: Yes

Reviewer #2: N/A

Response 2: We appreciate the reviewers positive support, we did indeed had an expert biostatistician analyzing our data. 

3. Have the authors made all data underlying the findings in their manuscript fully available?

Reviewer #1: Yes

Reviewer #2: Yes

Response 3: Indeed, all the data are available in the supplementary files. 

4. Is the manuscript presented in an intelligible fashion and written in standard English?

Reviewer #1: Yes

Reviewer #2: Yes

Response 4: Once again on behalf of the co-authors and myself we are grateful and appreciate the positive feedback. 

5. Review Comments to the Author

Reviewer #1: This is a very important manuscript. It highlights the general public knowledge on HIV/AIDS. most importantly its highlight the perception of health workers towards HIV patients. The attitude of healthcare workers needs more improvement to provide the best amount of care to the HIV-afflicted.

Response 4 for Reviewer 1: Indeed these data and our finding once published will share it with policy makers and the ministry of health to start awareness campaigns in the country especially among health care workers in order to provide the best healthcare to this group of patients without discrimination.

Reviewer #2: This is a good study, unfortunately dataset and results are not unique. The paper has many shortcomings in regard to data analyses and text. In my opinion, However, it needs to more clarify regarding to the protocol of sample collection and protocol which is used to online survey. In addition, results has not been used to its full text. I think need more in-depth analysis of the data. I also suggest siting more relevant and new literatures.

Response 4 for Reviewer 2: Valid point and accordingly, we have addressed this in the data collection method by describing how the online survey was disseminated through social media posts such as the twitter pots of our institute KFSHRC and WhatsApp platform of data collection volunteers. 

On behalf of the co-authors and myself we are grateful and appreciate your time in reviewing our project and we hope to collaborate on many more projects in the near future. 

Thank you

---

## [Decision Letter · Decision Letter 1]

20 Jun 2023

PONE-D-22-34156R1Assessing the Knowledge and Attitude Towards HIV/AIDS among the General population and Health Care Professionals in MENA regionPLOS ONE

Dear Dr. AlMozaini,

Thank you for submitting your manuscript to PLOS ONE. After careful consideration, we feel that it has merit but does not fully meet PLOS ONE’s publication criteria as it currently stands. Therefore, we invite you to submit a revised version of the manuscript that addresses the points raised during the review process.

ACADEMIC EDITOR: The authors need to address the following concerns of reviewer 3 to strengthen their manuscript:There is discrepancy in the statistical tool used to analysed in the data. In the abstract, R-statistical tool was mentioned, while SPSS was mentioned in the methods section of the manuscript. The author needs to address it.Some participants were less than 18 years. The consent of their parents/guardians are required before they can be interviewed. The authors were silent on this issue. The authors need to address this issue. ==============================

We look forward to receiving your revised manuscript.

Kind regards,

Frank Kyei-Arthur, Ph.D.

Academic Editor

PLOS ONE

Journal Requirements:

Additional Editor Comments:

The authors need to address the following concerns of reviewer 3:

• There is discrepancy in the statistical tool used to analysed in the data. In the abstract, R-statistical tool was mentioned, while SPSS was mentioned in the methods section of the manuscript. The author needs to address it.

• Some participants were less than 18 years. The consent of their parents/guardians are required before they can be interviewed. The authors were silent on this issue. The authors need to address this issue.

Reviewers' comments:

Reviewer's Responses to Questions

**Comments to the Author**

1. If the authors have adequately addressed your comments raised in a previous round of review and you feel that this manuscript is now acceptable for publication, you may indicate that here to bypass the “Comments to the Author” section, enter your conflict of interest statement in the “Confidential to Editor” section, and submit your "Accept" recommendation.

Reviewer #1: All comments have been addressed

Reviewer #3: (No Response)

2. Is the manuscript technically sound, and do the data support the conclusions?

Reviewer #1: Yes

Reviewer #3: Yes

3. Has the statistical analysis been performed appropriately and rigorously? 

Reviewer #1: Yes

Reviewer #3: Yes

4. Have the authors made all data underlying the findings in their manuscript fully available?

Reviewer #1: Yes

Reviewer #3: Yes

5. Is the manuscript presented in an intelligible fashion and written in standard English?

Reviewer #1: Yes

Reviewer #3: Yes

6. Review Comments to the Author

Reviewer #1: All comments have been answered adequately. The authors took into consideration all recommendations.

Reviewer #3: I reckon the flow of the article made it interesting to review. However, i was expecting analysis output from R software as stated on the abstract. Why was SPSS used instead. Also, how was consent gotten from teenagers who participated in the survey. Were their parents involved. Thank you. Great Article

7. PLOS authors have the option to publish the peer review history of their article (what does this mean?). If published, this will include your full peer review and any attached files.

Reviewer #1: No

Reviewer #3: **Yes: **Adeyemi, Adebowale Sylvester

While revising your submission, please upload your figure files to the Preflight Analysis and Conversion Engine (PACE) digital diagnostic tool, https://pacev2.apexcovantage.com/. PACE helps ensure that figures meet PLOS requirements. To use PACE, you must first register as a user. Registration is free. Then, login and navigate to the UPLOAD tab, where you will find detailed instructions on how to use the tool. If you encounter any issues or have any questions when using PACE, please email PLOS at figures@plos.org. Please note that Supporting Information files do not need this step.<quillbot-extension-portal></quillbot-extension-portal>

---

## [Author Response · Author response to Decision Letter 1]

27 Jun 2023

1. There is discrepancy in the statistical tool used to analysed in the data. In the abstract, R-statistical tool was mentioned, while SPSS was mentioned in the methods section of the manuscript. The author needs to address it.

Response: We thank the reviewers for noticing this minor mistake, we have corrected this by adding the R statistical tool that was utilized for our study. Red highlight in the manuscript in the statistical methodology section.

2. Some participants were less than 18 years. The consent of their parents/guardians are required before they can be interviewed. The authors were silent on this issue. The authors need to address this issue.

Response: On behalf of all the coauthors and myself, we appreciate this point, accordingly we have addressed this in the ethics section that is highlighted in red. This comment was added ‘Some participants were less than 18 years, 12% of our study participant. The consent of their parents/guardians were obtained before they were interviewed’.

---

## [Editor Report · Decision Letter 2]

6 Jul 2023

Assessing the Knowledge and Attitude Towards HIV/AIDS among the General population and Health Care Professionals in MENA region

PONE-D-22-34156R2

Dear Dr. AlMozaini,

We’re pleased to inform you that your manuscript has been judged scientifically suitable for publication and will be formally accepted for publication once it meets all outstanding technical requirements.

Kind regards,

Frank Kyei-Arthur, Ph.D.

Academic Editor

PLOS ONE

Additional Editor Comments (optional):

Reviewers' comments:

---

## [Editor Report · Acceptance letter]

19 Jul 2023

PONE-D-22-34156R2 

Assessing the Knowledge and Attitude Towards HIV/AIDS among the General population and Health Care Professionals in MENA region 

Dear Dr. Al-Mozaini:

I'm pleased to inform you that your manuscript has been deemed suitable for publication in PLOS ONE. Congratulations! Your manuscript is now with our production department. 

Kind regards, 

on behalf of

Dr. Frank Kyei-Arthur 

Academic Editor

PLOS ONE